# Long-Term Effects of Whole-Body Vibration on Hind Limb Muscles, Gait and Pain in Lame Dogs with Borderline-to-Severe Hip Dysplasia—A Pilot Study

**DOI:** 10.3390/ani13223456

**Published:** 2023-11-09

**Authors:** Mayara Viana Freire Gomes, Sheila Canevese Rahal, Ivan Felismino Charas dos Santos, Carmel Rezende Dadalto, Maria Jaqueline Mamprim, Roberta Rocha Negrão, Joel Mesa Hormaza, Miriam Harumi Tsunemi, Constanza B. Gómez Álvarez

**Affiliations:** 1School of Veterinary Medicine and Animal Science, São Paulo State University (UNESP), Botucatu 18618-681, Brazil; mayaravfgomes@gmail.com (M.V.F.G.); ivan.santos@unesp.br (I.F.C.d.S.); carmel_dadalto@outlook.com (C.R.D.); jaqueline.mamprim@unesp.br (M.J.M.); 2Department of Biophysics and Pharmacology, Bioscience Institute, São Paulo State University (UNESP), Botucatu 18618-689, Brazil; roberta.negrao@unesp.br (R.R.N.); joel.mesa@unesp.br (J.M.H.); 3Department of Biodiversity and Biostatistics, Bioscience Institute, São Paulo State University (UNESP), Botucatu 18618-689, Brazil; m.tsunemi@unesp.br; 4Department of Veterinary Medicine, University of Cambridge, Cambridge CB2 1TN, UK; 5Department of Life Sciences, Brunel University London, London UB8 3PH, UK

**Keywords:** canine lameness, hip dysplasia, muscle atrophy, whole-body vibration

## Abstract

**Simple Summary:**

Whole-Body Vibration (WBV) on vibrating platforms has been used as an alternative method of physiotherapy and rehabilitation for musculoskeletal, neurological or metabolic conditions in humans. However, in dogs, the use of WBV as a physical therapeutic modality has not been well established. This pilot study used several parameters to evaluate the long-term effects of WBV in lame dogs with borderline-to-severe hip dysplasia diagnosed radiographically. Although these results were preliminary, WBV significantly increased the size of both hind limb quadriceps femoris muscles and the left gluteal muscle. The owner’s perception was that, during the trial period, their dog’s pain decreased. However, no significant changes in the gait pattern or lameness score were found. Further studies evaluating the use of WBV for canine hip dysplasia appear to be justified.

**Abstract:**

This pilot study aimed to evaluate the long-term effects of Whole-Body Vibration (WBV) on hind limb muscles, gait and pain in lame dogs with borderline-to-severe hip dysplasia. Ten lame client-owned dogs with borderline-to-severe hip dysplasia, aged from 1.5 to 9.0 years and weighing 14.5 to 53.0 kg, were enrolled. The WBV training program consisted of 15 min sessions three times weekly for 16 weeks. Muscles of the hind limbs were evaluated using measurements of thigh circumference, the cross-sectional thickness of selected hind limb muscles by ultrasound assessment, and vastus lateralis muscle activity determined by surface electromyography (EMG). Lameness and clinical signs of pain were assessed by visual lameness scoring, orthopedic examination and an owner-based questionnaire. Kinetic analysis was performed by using a pressure-sensitive walkway. Manual thigh circumference measurements of both hind limbs showed significant increases over the trial period with a greater degree of change observed after week 8. Ultrasound measurements of the left gluteal muscles and the quadriceps femoris muscles of both hind limbs showed significant increases in the cross-section thickness post WBV. Owner’s perception of pain also showed a decrease in signs of pain at week 12 and week 16 compared to baseline. Based on graphs of the EMG activity patterns of the vastus lateralis muscle, 65% of the hind limbs had an improvement after 48 WBV sessions when compared to pre-session patterns. However, no significant differences were observed in visual lameness evaluation and kinetic analysis. Therefore, further studies will help to better clarify the role of WBV in canine rehabilitation protocols.

## 1. Introduction

Canine hip dysplasia is a complex developmental skeletal disease that has an apparently continuous progression over time [1,2]. The clinical signs of dysplasia may present in juvenile dogs related to tearing and inflammation of the joint capsule along with acetabular microfracture, and in adult dogs older than 12 months of age, associated with osteoarthritis progresses [2,3,4]. Even though radiographic signs of osteoarthritis may be visualized sometime after 4 to 6 months of age [2], the clinical presentation is very variable [3,4]. Hip pain, lameness, reduced locomotor activity, gait abnormalities and atrophy of thigh muscles may be observed in dogs with borderline-to-severe hip dysplasia [3,4,5,6].

Several non-surgical interventions are available for the management of canine hip dysplasia, such as pharmacological treatment, nutritional management, weight control, environmental modification, use of platelet-rich plasma or mesenchymal stem cells, exercise, physical rehabilitation, as well as combination of therapy approaches [7,8,9]. The goals of conservative management are to decrease or eliminate pain and muscle atrophy, decrease the disease progression and consequently improve or regain the normal activity [8,9,10].

Whole-Body Vibration (WBV) on vibrating platforms has been used for improving muscle performance in different types of sports and as an alternative method of physiotherapy and rehabilitation for musculoskeletal, neurological or metabolic conditions in humans [11,12]. The mechanism by which WBV may enhance performance and muscular strength has not been well understood [12,13]. Improvement in quadriceps muscle strength was observed after 8 weeks of WBV in women with knee osteoarthritis [14]. In addition, WBV use in human patients with knee osteoarthritis has showed therapeutic effects for reducing pain and improving joint function [15].

However, in dogs, the use of WBV as a physical therapeutic modality has not been well established. A few reports are related to the influence of therapy on different biological aspects in healthy dogs [16] and in dogs with hip dysplasia treated with reticulated hyaluronic acid alone or associated with WBV [17]. Therefore, this pilot study aimed to evaluate the long-term effects of WBV on hind limb muscles, gait and pain in lame dogs with borderline-to-severe hip dysplasia. The hypothesis proposed was that WBV provides a beneficial effect on improving muscle function with consequent improvement in clinical signs.

## 2. Materials and Methods

### 2.1. Dogs and Radiographic Examination

Twenty-eight medium-to-large-sized, adult, client-owned dogs with a history of hind limb lameness were evaluated. The inclusion criteria included hind limb lameness, and pain on palpation and/or crepitation of the hip joint. The exclusion criteria included dogs that had previous orthopedic surgery; undergoing drug pain management, joint supplement, or rehabilitation protocol during the last three months; presence of any type of neurological disease, being pregnant; presenting alterations in hematological and biochemical parameters; aggressive dogs; and owner’s unavailability to bring the dog to participate during the full WBV protocol.

Ventrodorsal hip-extended radiographic projections of the pelvis were obtained with a digital X-ray system (Digital X-ray system; GE Health DR-F, Beijing, China) prior to beginning the study. The dogs were sedated with intramuscular methadone (0.3 mg/kg) and acepromazine (0.05 mg/kg). Then, propofol was used intravenously (2 mg/kg). Dogs were positioned in dorsal recumbency with the hind limbs extended caudally and held parallel. Stifle joints were rotated medially, with the patella centered within the trochlear groove for each pelvic limb. The technique used was a focal-film distance of 100 cm, 60–90 kV, and 5.0–6.4 mAs. Following the Orthopedic Foundation for Animals classification [18], hip conformation was assigned scores ranging from 0 to 4 for each hip joint, with the following meanings: 0 representing not dysplastic categories (excellent, good, and fair), and 1, 2, 3, and 4 representing borderline, mild, moderate and severe categories, respectively. In addition, radiographic exams of the forelimbs were performed in dogs that showed signs of pain or discomfort upon manipulation during the orthopedic examination. Radiographs were interpreted by an experienced veterinary radiologist.

### 2.2. WBV

To perform WBV sessions, each dog was maintained in a standing position in the center of a tri-planar vibration platform of 92 cm length, 62 cm width and 16 cm height (TheraPlate Revolution^®^; Weatherford, TX, USA), in an acclimatized room at 22 °C. All dogs were submitted to the same protocol as previously described [16], which included a frequency of 30 Hz for 5 min (peak displacement of 3.10 mm; peak acceleration of 55.0 m·s^−1^), followed by 50 Hz for 5 min (peak displacement of 3.98 mm; peak acceleration of 195.96 m·s^−1^) and finishing with 30 Hz for 5 min (peak displacement of 3.10 mm; peak acceleration of 55.0 m·s^−1^). The frequency was previously checked using a Digital Oscilloscope (UNI-T UTD2102e^®^, Bangladesh, India). The acceleration was determined using a 3-axis digital accelerometer sensor (LSM6DSM Accelerometer^®^, STMicroelectronics, São Paulo, Brazil) placed in the center of the platform. WBV training program consisted of 15 min sessions 3 times weekly for 16 weeks (total = 48 sessions).

### 2.3. Evaluation of Body Condition

The body weight was assessed by a digital scale, and the body condition scoring was evaluated by a 9-point body condition scoring system [19]. The body mass index was determined by the formula: body weight (kg)/height (m)^2^. The height was measured from the atlanto-occipital joint passing through the spine line to the ground, immediately behind the hind limbs. All evaluations were conducted before WBV sessions and in week 4, week 8, week 12, and week 16 of the WBV protocol.

### 2.4. Pain and Lameness Assessments

An owner-based questionnaire was used to evaluate pain intensity based on one previously described for dogs with hip dysplasia and translated into Portuguese [20]. The questions answered by the owner’s dog had a descriptive scale of 0, 1, 2, 3 or 4. Each item was scored from 0 (the best) to 4 (the worst). A well-established lameness evaluation scoring system [21] was utilized at the walk and trot by an experienced veterinarian in orthopedics, described as follows: 0—walks/trots normally, 1—slight lameness; 2—obvious weight-bearing lameness, 3—severe weight-bearing lameness, 4—intermittent non-weight-bearing lameness, 5—continuous non-weight-bearing lameness. All dogs were filmed. At the same time, an orthopedic evaluation was also performed by an experienced veterinarian in orthopedics. Hip joint examination (pain, crepitus, decreased range of motion) was classified, as follows: 1—absent, 2—mild, 3—moderate, 4—severe. All evaluations were conducted before WBV and every four weeks during the WBV protocol.

### 2.5. Blood Tests and Muscle Size Evaluations

Blood tests included complete blood count (CBC) and creatine kinase. The measurements of the thigh circumference of both hind limbs were performed by using metric tape with the dogs in a weight-bearing position. After clipping the hair coat, the thigh circumference was measured midway between the greater trochanter and lateral femoral condyle. The measurements were performed in triplicate by one investigator. The mean was calculated from the 3 measurements taken and this mean value was used for the statistical calculations.

Ultrasound examination of the hind limb muscles was performed with a high-resolution ultrasound machine (MyLab Alpha, Esaote, Genoa, Italy) equipped with a multi-frequency linear transducer (3 MHz–13 MHz). Gluteal, sartorius, vastus, quadriceps femoris, and biceps femoris muscles were measured with dogs physically restrained in the lateral recumbency. Isopropyl alcohol and acoustic gel were used to avoid image artifacts. Measurements (cm) were carried out in transversal planes at the maximal thickness in each muscle. Images of cranial, medium, and caudal sections were taken in gluteal muscles using the greater trochanter of the femur and the ischiatic tuberosity as anatomical landmarks. For the other muscle groups, the anatomic points were the greater trochanter of the femur and proximal point of the patella, and the images were taken proximal, medium, and distal sections as previously described [22]. For quadriceps femoris muscle thickness, the measurement was performed exclusively in the midthigh, which passed through the center of the rectus femoris muscle. All pre- and post-treatment ultrasound measurements were taken by a single experienced investigator utilizing bony landmarks to ensure that the images were taken at the same location.

Laboratory tests and muscle evaluations were conducted before WBV and at weeks 4, 8, 12 and 16 during the WBV protocol.

### 2.6. Kinetic Gait Analysis

The kinetic analysis was carried out using a pressure-sensing walkway system (Walkway High-Resolution HRV4; Tekscan, South Boston, MA, USA). The sensors were equilibrated and calibrated as determined by the manufacturer. Each dog was conducted across the pressure-sensing walkway in a straight line, with controlled velocity (0.9–1.1 m/s) and acceleration (−0.15–0.15 m/s^2^). Approximately 20 trials at walk were collected for each dog, and the first five valid trials were selected. A valid trial included the four limbs in contact with the walkway surface during each gait cycle, without pulling on the leash or trying to turn the head. The percentage of body weight distribution among the four limbs was established using the formula: (peak vertical force of the limb/total peak vertical force of the four limbs) × 100.

Metacarpus/metatarsus pad, toe and paw contact areas (cm^2^) measurements were obtained from the pressure platform during stance phases in each valid trial and compared among evaluations for each dog.

The evaluations were conducted before WBV and at week 8 and week 16 of WBV sessions.

### 2.7. Surface Electromyography (EMG)

The surface *EMG* signals were recorded by using a surface EMG system coupled to an accelerometer (Trigno™ Wireless EMG Mini Sensor; Delsys, MA, USA). The right and left vastus lateralis muscles were recorded simultaneously and at the same time that the dogs walking on pressure-sensing walkway. After clipping and cleaning with alcohol the skin at the lateral area of the thigh, the sensor’s mini-head was placed by a single investigator, based on the previously described [23]. The distance between the iliac crest and the patella (line 1) and the distance between the patella and the major trochanter (line 2) were determined. The midpoints of lines 1 and 2 were established and connected by a third line. The EMG sensor was placed longitudinally approximately 1 cm from the midpoint of the third line. The surface EMG signals related to inertial sensors (accelerometry) were evaluated before WBV and at week 8 and week 16 of WBV protocol.

A sampling rate of 2000 Hz was used for surface EMG signals, and 148.1481 Hz for accelerometer signals, following the system manufacturer’s recommendations. The mean of five valid trials, as determined by gait analysis, was selected.

The processing of accelerometer signals adhered to the protocol established in reference [24], ensuring consistent and reliable data analysis. Acceleration was measured in units of gravity and sampled at a rate of 148.148 Hz for accelerometer signals, in accordance with the manufacturer’s recommendations. The raw accelerometer data from the tri-axial accelerometer was used without any preprocessing. Since the accelerometer was tri-axial, the individual axes alone were not suitable due to orientation issues. Therefore, it was necessary to calculate the total acceleration using the following expression:(1)$$a_{i}=\sqrt{{a_{xi}}^{2}+{a_{yi}}^{2}+{a_{zi}}^{2}}$$

Here, $a_{i}$ represents the total acceleration, and $a_{xi}$, $a_{yi}$, and $a_{zi}$ are the acceleration components along each coordinate axis provided by the accelerometer over time. This calculation was performed for each hind limb (right and left).

In addition to determining correlation and asymmetric functions, we employed an algorithm for step detection and identification of stance and swing phases in healthy dogs. For step detection, the magnitude of acceleration $a_{i}$ from Equation (1) was used to calculate the local mean acceleration $\bar{a_{i}}$ as follows:(2)$${\bar{a}}_{i}=\frac{1}{2\delta +1}\sum_{j=i-\delta }^{i+\delta }a_{j}$$
where ‘$a_{j}$’ represents acceleration values at time j, and δ is the signal averaging window. Subsequently, we computed local acceleration standard deviations $\sigma_{i}$ from the local mean acceleration ${\bar{a}}_{i}$ to highlight limb activity, using the following equation:(3)$${\sigma_{i}}^{2}=\frac{1}{2\delta +1}\sum_{j=i-\delta }^{i+\delta }{\left(a_{j}-{\bar{a}}_{j}\right)}^{2}$$

Stance phases were identified when the local acceleration standard deviation exceeded the first threshold, while swing phases commenced when it fell below the second threshold. In the absence of data collected from a treadmill with force plates, we determined the threshold values based on temporal coincidences between extreme points for each axis and total accelerometer signals (minimum, maximum and zero). We also obtained mean motion cycles from surface EMG data, with temporal variables for stance and swing phases in the activity pattern, as previously reported for the vastus lateralis muscle in dogs. In this case, accelerometer signals were filtered using a second-order, low-pass Butterworth filter with a cutoff frequency of 4 Hz to reduce noise. Considering both limbs, the upper threshold magnitude was established from the mean of interpolated values of standard deviations (defined in Equation (3)) calculated for time positions of maxima in the filtered *z*-axis component of the accelerometer. Conversely, the lower threshold was defined from the mean values of interpolated standard deviations for time positions where the accelerometer in the *z*-axis had a minimum. For all calculations, a value of δ = 15 was adopted.

As each gait cycle varies in duration, we resampled the signal to consist of 50 points and standardized it within a range of 0 to 100, using the maximum amplitude and time interval for normalization. The mean (±standard deviation) was computed from five valid gait cycles for each dog.

The surface EMG signals were evaluated using subjective criteria as follows:0: The graph in stance and swing phases closely resembled the normal pattern;1: The graph had a pattern similar to the normal one, but with pronounced peaks;2: One of the phases (stance or swing) did not match the normal pattern;3: Neither stance nor swing phases matched the normal pattern.

This methodology allowed for a structured analysis of the surface EMG signals.

### 2.8. Statistical Analyses

The assumption of normality of quantitative variables (body weight, body mass index, lameness score, pain score, blood tests, manual thigh circumference measurements, ultrasound measurements, kinetic data) was assessed by the Shapiro–Wilk test. When the assumption of normality was satisfied, the mixed models were adjusted with the moments as random effects. Otherwise, Friedman’s nonparametric test with Wilcoxon signed-rank post hoc test was applied to compare median among time points. Values of *p* < 0.05 were considered significant. Statistical analysis was performed with R Core Team, R: A Language and Environment for Statistical Computing. Descriptive statistics were expressed as Mean and Standard Deviation.

## 3. Results

### 3.1. Dogs, Radiographic Findings and Evaluation of Body Condition

A total of 10 dogs met the inclusion criteria (Table 1). Five dogs were intact (nos. 1, 2, 4, 5, 6), and five were spayed/neutered (nos. 3, 7, 8, 9, 10). The ages ranged from 1.5 to 9.0 years (Mean ± Standard Deviation: 5.5 years ± 3.06), and the body weight from 14.5 to 53.0 kg (33.27 kg ± 11.82). The protocol was well-tolerated by the dogs. A total of eight dogs had bilateral hip dysplasia. One dog was borderline in both hip joints (no. 7), and the other was borderline in the right hip joint (no. 6) (Table 2). Radiographs of the forelimbs were performed on one dog (no. 3) with pain on elbow palpation. Mild signs of osteoarthritis were verified.

The body condition scoring varied from 4.0 to 8.0 (mean 6.0) with a body mass index of 14.0–27.0 (mean 19.79 ± 4.25) (Table 1). No statistically significant differences were found in body weight (*p* = 0.87) and body mass index (*p* = 0.90) among all evaluation time points (Table 1).

### 3.2. Pain and Lameness Assessments

The subjective scores obtained by a questionnaire completed by 10 owners about signs of pain, discomfort, and difficulties with locomotion revealed a statistically significant decrease among evaluation time points (*p* = 0.03). The decrease in signs of pain according to owner-based questionnaire (Figure 1) was more than or equal to 25% at week 12 in four dogs (nos. 2, 5, 7, 10) and at week 16 in five dogs (nos. 2, 5, 7, 9, 10) when compared to week 0.

Except dog no. 6 that had unilateral lameness (right hind limb), all the other dogs showed had mild-to-severe bilateral hip dysplasia. In four dogs (nos. 1, 3, 4, 8), more severe left hind limb lameness was observed, and in six dogs (nos. 2, 5, 6, 7, 9, 10), right hind limb lameness occurred more severely (Table 2). One dog (no. 3) also showed discomfort on palpation of the elbows and radiographically had signs of osteoarthritis which was more accentuated in the left side. No significant differences were found for lameness score in each limb at both walking (forelimbs: *p* = 0.95; hind limbs: *p* = 0.85) and trotting (forelimbs: *p* = 0.56; hind limbs: *p* = 0.43) gaits among time points.

### 3.3. Blood Tests and Muscle Evaluations

The CBC did not present statistical differences among evaluation time points. The exception was platelet count that showed a statistically significant reduction between all evaluation time points (*p* = 0.04) but within the reference values. No significant differences were found in creatine kinase values among all evaluation time points (*p* = 0.76). Statistically significant differences (*p* < 0.01) were found in thigh circumference measurements in both hind limbs, which were more evident after week 8 of WBV (Figure 2). Ultrasound measurements of muscle thickness showed a statistically significant increase in the following muscles: cranial area of the left gluteal muscles (*p =* 0.04) between week 8 and week 16; caudal area of the left gluteal muscles (*p* = 0.01) between week 0, week 12 and week 16; middle area of the left quadriceps femoris muscles between week 0 and week 16; middle area of the right quadriceps femoris muscles (*p =* 0.01) between week 0 and week 16 (Figure 3). The values of the measurements of these muscles are in Table 3. No significant differences were seen in sartorius, vastus and biceps femoris muscles for all evaluation time points.

### 3.4. Kinetic Gait Analysis and Surface EMG

No significant differences were observed between dogs for the percentage of body weight distribution (Table 4) and the metacarpal/metatarsal pad, toes and paw contact areas of each limb (Table 5). The contact area between forelimbs and hind limbs was more symmetric between left and right hind limbs at week 16.

Before WBV session (week 0) and at week 8 (after 24 sessions of WBV), the percentage of muscle activity, as determined by surface EMG, in the vastus lateralis muscle showed a decrease during the stance phase. The peak in muscle activity occurred at the end of the stance phase, and variations in muscle activity were observed during the swing phase (Figure 4). At week 16 (after 48 sessions of WBV), there was an increase in the percentage of muscle activity, and the peak shifted to the middle third of the stance phase. Therefore, based on graphs of the EMG activity patterns of the vastus lateralis muscle, an improvement was verified after 48 WBV sessions when compared to pre-session patterns in both hind limbs for five dogs (nos. 1, 2, 4, 6, 7) and in the left hind limb for three dogs (nos. 3, 5, 9) (Table 6).

## 4. Discussion

Although these results were preliminary, data suggested that WBV provided a beneficial effect on some muscles and probably pain.

The vibration characteristics of the machine are one of the factors that may influence the effects of the WBV on the body [13]. For example, linear vibration plates produce vibrations that travel in a straight line, tri-planar vibration plates generate multidirectional vibrations, and pivotal vibration plates produce an oscillating motion [25]. The machine used in the present study (TheraPlate) differs from these because it provides vortex wave circulation that allows the pulsatile vibration to be equally distributed to all four limbs, which is well-tolerated by the dogs [17].

The total number of sessions used in the present study was higher than that reported by other studies in dogs using the TheraPlate [16,17]. However, the vibration platform used in those studies as well as the frequency and duration of each WBV were the same [16,17]. Vibration frequency and amplitudes as well as the number of sessions are relevant factors and should be set according to the WBV purpose [12,26], but a safety protocol must be used [16]. No adverse effect was detected in the dogs after WBV sessions, and no significant differences were found in CBC and creatine kinase levels for all evaluation time points. Therefore, the protocol used was considered safe and feasible for the objective of this study.

Radiographic findings, lameness score and orthopedic evaluation showed no association in 75% of the dogs in this study. Inconsistent relationships between radiographic features and clinical signs in dogs with hip dysplasia have been frequently reported [1,27]. In addition, some dogs do not exhibit clinical signs until the development of degenerative joint disease [1,4]. The dogs in the present study showed predominantly hind limb lameness at visual lameness scoring; however, signs of pain were observed in the forelimbs in one dog (no. 3). A study in dogs with hip dysplasia also reported signs of pain in other joints of the hind limbs as well as forelimb joints [7]. In addition, it is worth mentioning that the body weight and the body mass index did not influence the lameness score since no significant change occurred for all evaluation time points.

Significant changes were found in the thigh circumference measurements and ultrasound measurements of the left gluteal muscles and bilateral quadriceps femoris muscles after WBV in the dogs. The gluteal muscles are lateral muscles of the pelvis related to flexion and extension of the hip and abduction of the hind limb, while the quadriceps femoris muscle is a cranial muscle of the thigh used to support weight and an extensor of the stifle while also flexing the hip [28]. More studies are needed to ascertain the response in each thigh muscle since Whole-Body Vibration revealed statistically significant positive effects in two muscles but no difference in sartorius, vastus, and biceps femoris muscles on ultrasound estimation.

In addition, human studies have shown that WBV reduced atrophy in the knee and ankle extensors muscles in patients who underwent 56 days of bed rest [29]. In contrast, a randomized clinical study with healthy adult men showed no alteration of muscle activity or muscle architecture after 6 weeks of WBV training [30]. On the other hand, a systematic review of the use of WBV in the aging population identified nine studies that observed statistically significant improvements in lower-body muscular strength [31]. Based on these human studies, dogs with hip dysplasia that present low levels of fitness, and muscle wastage will be the most likely to obtain a beneficial effect with WBV.

Observational gait analysis using visual scoring scales has been used to validate the pain and lameness degree in dogs [20,32,33], since instrumented analysis, such as kinetic and kinematic analyses are not routinely used in clinical practice. The relationship between lameness score and kinetic evaluation has shown a good correlation in some studies [32,33,34]. However, the small sample size, the absence of a control group, the bilateral lesion, and the varying degrees of severity of the lameness in each hind limb may have negatively affected the ability to identify statistically significant differences in kinematic parameters in this study.

On the other hand, the scores obtained by a questionnaire completed by owners about signs of pain, discomfort, and difficulties with locomotion revealed a statistically significant decrease among evaluation time points. However, a study showed that interpreting owner reports of patient response to treatment for osteoarthritis might be influenced by the placebo effect [35]. For example, owners may perceive improvement in the dogs due to other issues than just an improvement in lameness, such as the dogs feeling happier and more comfortable. While these characteristics are subjective, they can reflect an improvement in the quality of life of these dogs due to the treatment.

The percentage of body weight distribution in each forelimb (>31.81%) was higher than each hind limb (<18.69%) for all evaluated time points. The percentage of body weight distribution in healthy dogs has been described as 30% on each forelimb and 20% on each hind limb [36,37,38], because of the location of the dog’s center of gravity near the forelimbs [39]. Dogs with lameness in hind limbs may have load redistribution to the forelimbs [40,41], as observed in the present study.

The mean paw contact area at the forelimbs (26.68 ± 8.66 cm^2^) was higher than at the hind limbs (20.62 ± 6.43 cm^2^). Similar findings were found in a study with healthy large dogs with mean paw area contact of 25.73 cm^2^ (±2.55) and 19.34 cm^2^ (±2.51), respectively for the paws of the forelimbs and hind limbs [36]. It was observed that the paw contact area was smaller for the lame limb than for the contralateral limb, however, with no significant difference. Similarly, one study found that alterations of limb loading are associated with changes in paw contact area [34]. Therefore, a decreased loading of the affected limb can result in decreased contact area values which was also found in this study.

The surface EMG signals from the vastus lateralis muscle showed different patterns during WBV. However, based on graphs of the EMG activity patterns of the vastus lateralis muscle, 65% of the hind limbs had an improvement after 48 WBV sessions when compared to pre-session patterns. The muscle activity during the stance phase and swing phase resembled those detected in a study that evaluated the vastus lateralis muscle before and after reversible moderate supporting lameness in healthy Beagle dogs [42]. Thus, the data before WBV and after 24 sessions of WBV (week 8) were near to the pattern observed on moderate supporting lameness, and the data after 48 sessions of WBV (week 16) were near to the pattern without lameness. In addition, two studies with large healthy dogs demonstrated peak activity in the stance phase and increase in the activity at the end of the swing phase of the vastus lateralis muscle [23,43], similar to the pattern of activity observed after 48 sessions of WBV (week 16) in 5 of 10 dogs evaluated in the present study. Therefore, it appears that the EMG signal showed a change in pattern from those observed in lame dogs to patterns observed in sound dogs in 50% of the dogs after the use of WBV for 16 weeks.

This pilot study has several limitations. First, the absence of a control group; therefore, the results should not be generalized. Unfortunately, owners of healthy dogs did not agree to participate in the study. Second, different dog breeds and small sample size were some inherent limitations associated with a clinical trial. Thus, further studies using a control group, specific breeds with similar radiographic and clinical signs, and a higher number of dogs are mandatory to confirm the role of WBV in hip dysplasia dogs. In addition, studies using other methodologies will be necessary to understand the role of WBV in pain reduction and/or inflammation and the different pain mechanisms involved in juvenile and mature dogs.

## 5. Conclusions

The findings of this pilot study in dogs with borderline-to-severe hip dysplasia suggested that a long-term WBV program could potentially improve hind limb muscle hypotrophy in some muscles and owner-reported pain in lame dogs. However, no significant changes were found in the percentage of limb loadings or for the visual lameness scores when pre-treatment evaluations were compared to post-treatment evaluations. Further studies evaluating the use of WBV for canine hip dysplasia appears to be justified.

## Figures and Tables

**Figure 1 animals-13-03456-f001:**
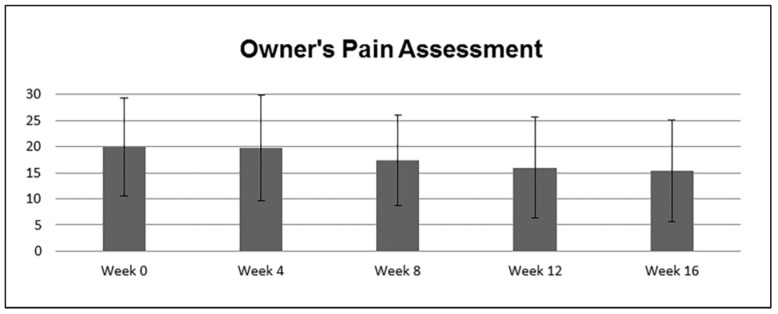
Box plot of the pain scores obtained by a questionnaire completed by owners before the Whole-Body Vibration (week 0), and 4, 8, 12 and at 16 weeks of Whole-Body Vibration in 10 lame dogs with borderline to severe hip dysplasia.

**Figure 2 animals-13-03456-f002:**
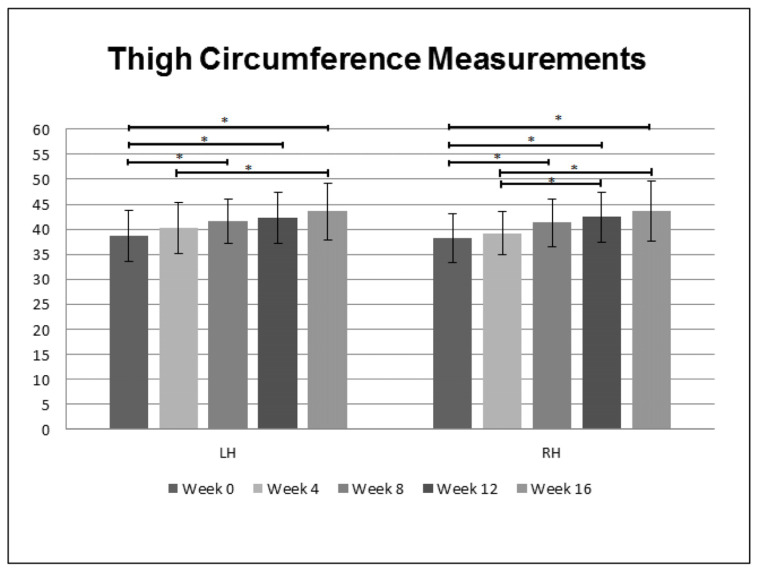
Box plot of the manual thigh circumference measurements (cm) in left hind limb (LH) and right hind limb (RH) obtained before (week 0), and at 4, 8, 12 and 16 weeks of Whole-Body Vibration in 10 lame dogs with borderline to severe hip dysplasia. Asterisk (*) indicates significant difference (*p* < 0.05) among evaluation time points.

**Figure 3 animals-13-03456-f003:**
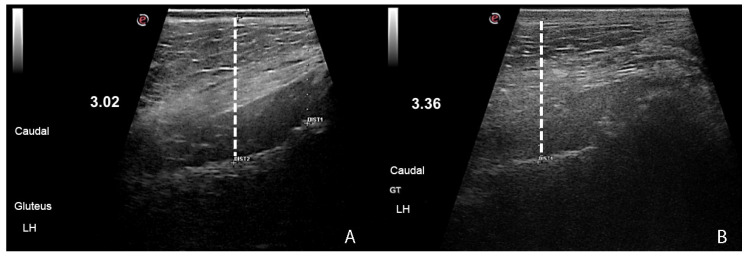
Ultrasound images of the gluteal muscles, obtained in the caudal region of the left hind limb of a dog with hip dysplasia, before (**A**) the Whole-Body Vibration (week 0) and (**B**) at week 16 of Whole-Body Vibration. Observe the increase in muscle thickness at week 16 (**B**).

**Figure 4 animals-13-03456-f004:**
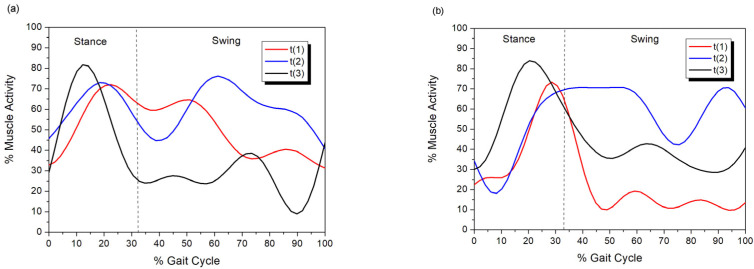
Surface EMG signs of muscle activity of the vastus lateralis muscle ((**a**) left hind limb; (**b**) right hind limb) during stance phase and swing phase in a dog (no. 4) with hip dysplasia, before (t1) and at weeks 8 (t2) and 16 (t3) of Whole-Body Vibration.

**Table 1 animals-13-03456-t001:** Dog assignment, body weight, 9-body condition score (BCS), and body mass index (BMI) in 10 lame dogs with borderline to severe hip dysplasia before the Whole-Body Vibration exercises.

Number	Dog Assignment (Breed, Approximate Age, Sex)	Body Weight (kg)	BSC	BMI
1	Crossbreed, 1.5-year-old, female	20.5	4	15.50
2	Rottweiler, 2-year-old, female	33.5	6	18.65
3	Crossbreed, 8-year-old, female	42.5	7	22.98
4	Labrador retriever, 9-year-old, male	38.5	8	26.74
5	German Shepherd, 8-year-old, male	39.4	5	15.39
6	Crossbreed, 8-year-old, female	53.2	8	24.62
7	Crossbreed, 2-year-old, female	23.4	6	18.99
8	Crossbreed, 3-year-old, female	14.5	4	13.53
9	Crossbreed, 6-year-old, male	40.6	6	21.04
10	Crossbreed, 8-year-old, female	20.6	8	20.47

**Table 2 animals-13-03456-t002:** Radiographic hip score according to Orthopedic Foundation for Animals (OFA) scores, lameness score at the walk, and orthopedic examination of the right (RH) and left (LH) hind limbs in 10 lame dogs with borderline to severe hip dysplasia before Whole-Body Vibration.

	Radiographic Hip Score		Lameness Score		Orthopedic Examination	
Dog	LH	RH	LH	RH	LH	RH
1	2	2	2	1	2	1
2	3	3	2	3	2	3
3	4	4	3	2	3	3
4	2	2	3	2	2	2
5	4	4	2	3	3	3
6	1	2	0	2	1	3
7	1	1	2	3	2	2
8	4	4	4	3	4	4
9	4	4	2	3	*	*
10	4	4	1	3	1	3

* dogs did not tolerate manual evaluation. Radiographic hip score: 0 = no; dysplastic categories, 1 = borderline, 2 = mild, 3 = moderate, 4 = severe. Lameness score: 0 = no; lameness with weight-bearing, 1 = slight; 2 = obvious; 3 = severe; lameness non-weight-bearing, 4 = intermittent. 5—continuous. Orthopedic examination (pain, crepitus, decreased range of motion in the hip joint): 1 = absent, 2 = mild, 3 = moderate, 4 = severe.

**Table 3 animals-13-03456-t003:** Comparison of the ultrasonographic measurements (cm) of the left gluteal muscles and right and left quadriceps femoris muscles before (week 0), and at weeks 8 and 16 of Whole-Body Vibration exercises in 10 lame dogs with borderline to severe hip dysplasia by Friedmann’s test.

Muscles	Measurement Area	Week 0 Median (Q1–Q3)	Week 4 Median (Q1–Q3)	Week 8 Median (Q1–Q3)	Week 12 Median (Q1–Q3)	Week 16 Median (Q1–Q3)	*p* Value
*Gluteal* (left hind limb)	Cranial	2.12, (1.82–2.38)	2.110, (1.81–2.38)	2.195, (1.79–2.42)	2.14, (2.00–2.21)	2.520, (2.50–2.57)	0.045
	Middle	2.27, (2.10–2.64)	2.360, (2.15–2.57)	2.435, (1.99–2.55)	2.37, (2.14–2.42)	2.575, (2.30–2.81)	0.171
	Caudal	2.38, (1.75–2.64) ^a^	2.285, (1.99–2.57) ^ab^	2.515, (2.34–2.68) ^ab^	2.37, (2.04–2.56) ^a^	2.865, (2.52–3.08) ^b^	0.015
*Quadriceps femoris* (left hind limb)	Middle	2.96, (2.65–3.27) ^a^	3.120, (2.87–3.45) ^ab^	3.170, (3.03–3.57) ^ab^	3.34, (2.73–3.50) ^ab^	3.560, (3.14–3.69) ^b^	0.036
*Quadriceps femoris* right hind limb)	Middle	2.57, (2.99–3.17) ^a^	3.075, (2.71–3.44) ^ab^	3.230, (3.07–3.35) ^ab^	3.24, (2.75–3.50) ^ab^	3.45, (3.31–3.77) ^b^	0.001

Different letters in the line indicate statistically significant differences (*p* < 0.05).

**Table 4 animals-13-03456-t004:** Comparison of the percentage of body weight distribution in the forelimbs and hind limbs before (week 0), and at weeks 8 and 16 of Whole-Body Vibration exercises in 10 lame dogs with borderline to severe hip dysplasia by mixed model.

Limbs	Week 0 (Mean ± SD)	Week 8 (Mean ± SD)	Week 16 (Mean ± SD)	*p* Value
Left forelimb	31.81 ± 1.11	32.14 ± 2.65	32.72 ± 2.46	0.572
Right forelimb	32.12 ± 3.16	31.82 ± 3.45	32.12 ± 2.97	0.969
Left hind limb	17.38 ± 1.67	17.90 ± 2.51	16.90 ± 2.63	0.599
Right hind limb	18.69 ± 2.14	18.14 ± 2.22	18.26 ± 2.27	0.829

**Table 5 animals-13-03456-t005:** Comparison of paws, metacarpus/metatarsus pads and toes contact areas in the forelimbs and hind limbs before (week 0), and at weeks 8 and 16 of Whole-Body Vibration exercises in 10 lame dogs with borderline to severe hip dysplasia by mixed model.

Contact Area	Week 0 (Mean ± SD)	Week 8 (Mean ± SD)	Semana 16 (Mean ± SD)	*p* Value
Paws (cm^2^)				
Left forelimb	26.89 ± 7.80	27.84 ± 8.34	27.54 ± 8.72	0.963
Right forelimb	28.53 ± 8.97	27.80 ± 8.43	27.19 ± 8.73	0.936
Left hind limb	20.06 ± 5.78	20.17 ± 5.83	20.10 ± 5.96	0.999
Right hind limb	22.37 ± 7.11	21.61 ± 5.58	21.08 ± 5.86	0.887
Pads (metacarpus/metatarsus) (cm^2^)				
Left forelimb	9.51 ± 3.43	9.78 ± 3.39	9.73 ± 3.78	0.981
Right forelimb	9.60 ± 3.86	9.60 ± 3.74	9.50 ± 4.13	0.998
Left hind limb	4.66 ± 1.72	4.74 ± 2.01	4.61 ± 1.74	0.985
Right hind limb	5.76 ± 2.39	5.61 ± 2.04	5.18 ± 2.09	0.810
Toes (cm^2^)				
Left forelimb	17.39 ± 4.70	18.06 ± 5.09	17.82 ± 5.14	0.949
Right forelimb	18.93 ± 5.26	18.20 ± 4.87	17.67 ± 4.88	0.841
Left hind limb	15.77 ± 4.27	15.82 ± 4.18	15.83 ± 4.37	0.999
Right hind limb	16.61 ± 5.01	16.01 ± 3.75	15.90 ± 4.13	0.887

**Table 6 animals-13-03456-t006:** Subjective evaluation of the surface EMG activity patterns before (week 0) and at weeks 8 and 16 of Whole-Body Vibration exercises in 10 lame dogs with borderline to severe hip dysplasia.

	Week 0		Week 8		Week 16	
Dog Number	Left Hind Limb	Right Hind Limb	Left Hind Limb	Right Hind Limb	Left Hind Limb	Right Hind Limb
1	1	3	2	2	0	1
2	3	3	1	NE	2	2
3	3	2	2	3	2	2
4	3	2	2	3	2	1
5	3	3	3	2	2	3
6	3	3	3	3	1	2
7	3	3	3	2	2	2
8	3	2	2	3	3	2
9	3	3	3	2	2	3
10	2	2	3	3	3	3

NE: Not evaluated (insufficient quality for use of the surface EMG signals). (0: normal; 1: pattern similar to normal but with pronounced peaks; 2: stance or swing did not match normal; 3: stance and swing did not match normal).

## Data Availability

The data presented in this study are openly available in [Repositorio Institucional UNESP] at https://repositorio.unesp.br/items/cc5b944b-50c1-4703-bf37-c2bcaa1b5fc7.

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
