# Peer review of "Long-Term Effects of Whole-Body Vibration on Hind Limb Muscles, Gait and Pain in Lame Dogs with Borderline-to-Severe Hip Dysplasia—A Pilot Study"

_animals, 2023, doi:10.3390/ani13223456_

Round 1
Reviewer 1 Report
Comments and Suggestions for Authors
The study “Long-Term Effects of Whole-Body Vibration on Hind Limb Muscles, Gait and Pain in Lame Dogs with Hip Dysplasia” describes a rehabilitation treatment modality to manage canine hip dysplasia. Studies similar to the present one are required, as many treatment modalities are currently used in veterinary medicine, and specific treatment parameters/frequencies have not been extensively evaluated.
I find the study to be interesting and, overall, well-written. The evaluation modalities used are interesting and encompass different OA/OA-pain experience dimensions. However, I consider a few points that hinder the study’s overall quality. The major is the lack of a control group, which is essential to make conclusions. Perhaps this should be considered a case series, not a full study. You should avoid using book chapters as references, as they are not peer-reviewed. You should be able to find papers covering the topics in which you cited books.
Please find specific comments below.
Simple Summary:
“Long-term WBV program improved hind limb muscles atrophy and owner’s perception of pain in lame dogs with hip dysplasia, however, this did not reflect on changes in limb loadings or lameness score”. It would be best if you avoided statements like this one. You do not have a control group, so you cannot definitely conclude that the program was the reason for the improvement.
Introduction:
Line 58: “linear progression”. Should it be a continuous progression? It is not usually linear; it will improve and worsen with time, with a progressive worsening of the overall condition.
M&M:
More information is required or needs to be clarified. Did you have ethical approval for this study? Was there a confirmation of hip dysplasia? Was there a minimal age for inclusion? How many dogs were included/excluded, and why? Were dogs with other orthopedic diseases included? Was supplementation allowed? Or rehab protocols? Were the dogs sedated for the radiographic examination? With what? Who performed the classification? Did you measure height in m2? Which specific questionnaire was used to evaluate pain? Is it validated? Has it been tested for the particular language being used? Is the methodology for thigh circumference validated (please see https://doi.org/10.3389/fvets.2018.00203)?
Results:
OFA scores are usually given for the combined hips. You did not evaluate hips individually post-treatment. Why are results presented separately?
I may have missed it, but where are the post-treatment lameness and ortho examination scores?
Discussion:
Overall, the discussion should be re-adjusted based on the presented comments. The lack of a control group is a major limitation (that the authors don’t even acknowledge), and results should be interpreted with care and this in mind.
Comments on the Quality of English Language
Some improvements are required throughout the manuscript.
Author Response
REVIEWER 1
Thank you very much for taking the time and effort to review the manuscript. We sincerely appreciate your positive and constructive feedback.
Comments and Suggestions for Authors
The study “Long-Term Effects of Whole-Body Vibration on Hind Limb Muscles, Gait and Pain in Lame Dogs with Hip Dysplasia” describes a rehabilitation treatment modality to manage canine hip dysplasia. Studies similar to the present one are required, as many treatment modalities are currently used in veterinary medicine, and specific treatment parameters/frequencies have not been extensively evaluated.
I find the study to be interesting and, overall, well-written. The evaluation modalities used are interesting and encompass different OA/OA-pain experience dimensions. However, I consider a few points that hinder the study’s overall quality. The major is the lack of a control group, which is essential to make conclusions. Perhaps this should be considered a case series, not a full study. You should avoid using book chapters as references, as they are not peer-reviewed. You should be able to find papers covering the topics in which you cited books.
Answer: Thanks for your observation. The lack of a control group was included in the limitations of the study. We had used books that we considered of high quality. In addition, the journal rules did not forbid the use of books.
Please find specific comments below.
Simple Summary:
“Long-term WBV program improved hind limb muscles atrophy and owner’s perception of pain in lame dogs with hip dysplasia, however, this did not reflect on changes in limb loadings or lameness score”. It would be best if you avoided statements like this one. You do not have a control group, so you cannot definitely conclude that the program was the reason for the improvement.
Answer: The text was removed. The summary was changed according suggestions reviewer 2.
Introduction:
Line 58: “linear progression”. Should it be a continuous progression? It is not usually linear; it will improve and worsen with time, with a progressive worsening of the overall condition.
Answer: The text was changed as suggested: “…..dysplasia is a complex developmental skeletal disease that has an apparently continuous progression over time.”
M&M:
- More information is required or needs to be clarified. Did you have ethical approval for this study?
Answer: Yes. This part had been included in Institutional Review Board Statement, according to author instruction.
“Institutional Review Board Statement: The animal study protocol was approved by the Ethics Committee for the Use of Animals from the School of Veterinary Medicine and Animal Science, Unesp, Botucatu campus (CEUA: 125/2016).”
Informed Consent Statement:
Written informed consent from the owners of the dogs were obtained before the beginning of the study.
- Was there a confirmation of hip dysplasia?
Answer: Yes. It was described in 2.2. Radiographic examination
“Ventrodorsal hip-extended radiographic projections of the pelvis were obtained with a digital X-ray system (Digital X-ray system; GE Health DR-F, USA) prior to beginning the study.
Following the Orthopedic Foundation for Animals classification [18] for hip conformation was attributed scores of 0 to 4 for each hip joint, as follows: 0 representing not dysplastic categories (excellent, good, and fair), and 1, 2, 3, and 4 representing borderline, mild, moderate and severe categories, respectively. In addition, radiographic exams of the forelimbs were performed in dogs that showed signs of pain or discomfort upon manipulation during the orthopedic examination.”
- Was there a minimal age for inclusion?
Answer: No. “The inclusion criteria included hind limb lameness, and pain on palpation and/or crepitation of the hip joint.”
- How many dogs were included/excluded, and why?
Answer: As described Material and Methods: Twenty-eight medium to large-sized, adult, client-owned dogs with a history of hind limb lameness were evaluated.
The exclusion criteria included dogs that had previous orthopedic surgery; undergoing drug pain management, joint supplement, or rehabilitation protocol during the last three months; presence of any type of neurological disease, being pregnant; presenting alterations in hematological and biochemical parameters; aggressive dogs; and owner’s unavailability to bring the dog to participate during the full WBV protocol.
As described in results:
A total of 10 dogs met the inclusion criteria based on described above.
- Were dogs with other orthopedic diseases included? Was supplementation allowed? Or rehab protocols?
Answer: No. We included in the exclusion criteria to clarify: ………….The exclusion criteria included dogs that had previous orthopedic surgery; undergoing drug pain management, joint supplement, or rehabilitation protocol during the last three ……….”
- Were the dogs sedated for the radiographic examination? With what?
Answer: The information was included: “The dogs were sedated with intramuscular methadone (0.3 mg/kg) and acepromazine (0.05 mg/kg). Then, propofol was used intravenously (2 mg/kg).”
- Who performed the classification?
Answer: The information was included as required: “Radiographs were interpreted by an experienced veterinary radiologist.“
- Did you measure height in m2?
Answer: No, this is part of the formula.
The height was measured from the atlanto-occipital joint passing through the spine line to the ground, immediately behind the hind limbs.
- Which specific questionnaire was used to evaluate pain?
Answer: As described: Owner-based questionnaires were used to evaluate pain intensity, as previously described for dogs with hip dysplasia [20].
Hielm-Björkman, A.K.; Kuusela, E.; Liman, A.; Markkola, A.; Saarto, E.; Huttunen, P.; Leppäluoto, J.; Tulamo, R.M.; Raekallio, M. Evaluation of methods for assessment of pain associated with chronic osteoarthritis in dogs. J. Am. Vet. Med. Assoc. 2003, 222, 1552–1558. doi: 10.2460/javma.2003.222.1552
Is it validated? Has it been tested for the particular language being used?
Answer: Unfortunately no.
- Is the methodology for thigh circumference validated (please see https://doi.org/10.3389/fvets.2018.00203)?
Answer: Unfortunately we did not use this methodology.
Results:
- OFA scores are usually given for the combined hips. You did not evaluate hips individually post-treatment. Why are results presented separately?
Answer: The hips were not evaluated post-treatment. The results were separated because the data could influence the kinematic data.
- I may have missed it, but where are the post-treatment lameness and ortho examination scores?
Answer: These data had been included: “No significant differences were found for lameness score in each limb at both walking (forelimbs: p = 0.95; hind limbs: p = 0.85) and trotting (forelimbs: p = 0.56; hind limbs: p = 0.43) gaits among time points. The subjective scores obtained by a questionnaire completed by 10 owners about signs of pain, discomfort, and difficulties with locomotion revealed a statistically significant decrease among evaluation time points (p = 0.03). The decrease of signs of pain according to owner-based questionnaires (Figure 1) was more than or equal to 25% at week 12 in four dogs (nos. 2, 5, 7, 10) and at week 16 in five dogs (nos. 2, 5, 7, 9 e 10) when compared to week 0.”
Discussion:
Overall, the discussion should be re-adjusted based on the presented comments. The lack of a control group is a major limitation (that the authors don’t even acknowledge), and results should be interpreted with care and this in mind.
Answer: The text was included as suggested: “
“This pilot study has several limitations. First, the absence of a control group; therefore, the results cannot be generalized. However, owners of healthy dogs did not agree to participate in the study. Second, different dog breeds and sample size are some inherent limitations to a clinical trial. Thus, further studies using a control group, specific breeds with similar radiographic and clinical signs, and a higher number of dogs should be done.”
Comments on the Quality of English Language
Some improvements are required throughout the manuscript.
Answer: Improvements were done according other reviewer’s suggestions.
Reviewer 2 Report
Comments and Suggestions for Authors
Rework of discussion and results will greatly improve paper.

a few minor changes in clarity of explanations and grammar needed
Author Response
REVIEWER 2
Thank you very much for taking the time and effort to review the manuscript. We sincerely appreciate your positive and constructive feedback.
This study appears to be a pilot study and should reflect that in name.
Answer: The title was changed as suggested: “Long-Term Effects of Whole-Body Vibration on Hind Limb Muscles, Gait and Pain in Lame Dogs with Hip Dysplasia or Borderline – A Pilot Study”
One of the biggest issues is that there is no control group to prove that these results are actual treatment effects. You need to keep that rather large limitation in mind throughout the paper. Also, hip dysplasia was radiographically only boardline to mild in 4 of 10 dogs in this study, which seems that radiographs would be the best “gold” standard for diagnosing dysplasia. The assumption would be that radiographically the dysplasia would need to be moderate to sever to be clinically relevant. If this is not true, then you need to utilize research findings to support your use of “boardline” and “mild” as radiographically relevant to diagnosis the dog with dysplasia. You are claiming that this treatment effects dogs with dysplasia so you need to convince the reader that the dogs actually do have dysplasia.
Answer: The absence of a control group was included in the limitations of the study as suggested. Unfortunately, we could not find owners of normal dogs to participate of the study. The radiographic findings may not be correlated to the clinical signs in hip dysplasia. Some dogs with significant signs of hip dysplasia may not exhibit any clinical signs. Thus, the inclusion criteria in this study included hind limb lameness, and pain on palpation and/or crepitation of the hip joint. Due to the criticism the title and aim was changed:” Long-Term Effects of Whole-Body Vibration on Hind Limb Muscles, Gait and Pain in Lame Dogs with Hip Dysplasia or Borderline – A Pilot Study”
The other issue is that the radiographic and lameness/orthopedic examination results (Table 2) do not appear to be consistently proving that the dogs have, or do not have, dysplasia. You state in your discussion (line 290) that inconsistent results in these 3 areas are common which leads me to again question if some of these dogs actually have what would be considered diagnosed dysplasia. Did you do a statistical analysis of the difference between the radiographic, lameness and orthopedic results (even though the sample size is small)? Perhaps this could offer more convincing evidence that these dogs all have dysplasia. However, you do say this inconsistency in diagnostic methods is common so why did you use these 3 evaluators?
If you offer convincing evidence that the subjects were all the same, then the results would be more convincingly due to the treatment and not due to the variation in subjects. This big issue of consistency (that all your subjects have dysplasia) needs to be addressed primarily for you to get the reader to find validity in your results.
Answer: The inclusion criteria in this study included hind limb lameness, and pain on palpation and/or crepitation of the hip joint. If you observe in table only the left hip of the dog 6 is questionable, since the lameness score was 0. However, the title and aim were changed to include the borderline dog.
To me the results of this study do not convince me that the treatment effect is what you say it is. Especially when you use words like “can” (line 377 for example) and “improved”. The results of this type of study, especially without a control group need to be “may” and “potentially could improve”. There is also value in showing that the materiais and methods could be improved from what you decided to do because deficiencies are sometimes only evident after you have completed a study. Don’t hesitate to explain in detail what you could have done differently and why.
Answer: The limitations of the study were included as suggested.
The verb was also changes as suggested: “The findings of the present study suggested that the long-term WBV program could potentially improve hind limb muscle hypotrophy and pain as reported by the owners in lame dogs with hip dysplasia or borderline but with no change in percentage of limb loadings or visual lameness score.”
You need to use bony landmarks to utilize pre and post ultrasounds to report that a change in muscle thickness occurred or didn’t occur over time. I am not an ultrasonographer, but your images should be able to convince me that the measurements were taken at the same location pre and post. Make this clearer to the reader. I did not see the results of the ultrasonographic measurements in your paper but you state that there were significant findings so show the numbers in a Table. And also explain why these two measurement techniques lead to different significant results.
Answer: The text was changed to clarify: Measurements (cm) were carried out in transversal planes at the maximal thickness in each muscle. Images of cranial, medium, and caudal sections were taken in gluteal muscles using the greater trochanter of the femur and the ischiatic tuberosity as anatomical points. For the other muscle groups, the anatomic points were the greater trochanter of the femur and proximal point of the patella, and the images were taken proximal, medium, and distal sections as previously described [22]. For quadriceps femoris muscle thickness, the measurement was done exclusively in the midthigh, which passed through the center of the rectus femoris muscle. All ultrasound measurements in the different evaluation time points were undertaken at the same locations by a single experienced investigator.
You state that the EMG scores of the vasus lateralis muscles were “close to normal” was there a significant difference, if not then you need to state why you claim this. The reader needs actual numbers to support your statement.
Answer: The information about surface EMG was changed to clarify.
Your lack of significant difference in the kinetic analysis is important and needs to be clearly discussed. Just because muscles get bigger doesn’t mean that improves the welfare of the dog. Clearly acknowledge the limitations of your study which ultimately assists in the design of the next study.
Answer: The limitations were included.
Line 9 need affiliation of Alvarez
Answer: I apologize but my PDF the affiliation is included:
Department of Veterinary Medicine, University of Cambridge, and Department of Life Sciences, Brunel University London, United Kingdom
Line 12. Used as an exercise modality for improving muscle performance - is too generalized
Used as an alternative method specifically - specifically for what in rehabilitation
Answer: The text was changed to clarify: Whole-body Vibration (WBV) on vibrating platforms has been used as exercise modality for improving muscle performance in different types of sports and as an alternative method of physiotherapy and rehabilitation for musculoskeletal, neurological or metabolic conditions in humans
Line 14-18 This pilot study used several parameters to evaluate the with boarderline to sever hip dysplasia diagnosed radiographically. Although these results were preliminary, WBV significantly increased the size of both hind limb quadricep muscles and the left gluteal muscle. The owner’s perception was that, during the trial period, their dog’s pain decreased. However, no significant changes in the gait pattern or lameness score were found. Further studies evaluating the use of WBV for canine hip dysplasia appear to be justified.
Answer: Thanks. The text was changed as suggested.
Line 23 thickness of selected hind limb muscles
Answer: It was included.
Line 24 and hind limb muscle activity determined by surface....
Answer: It was included.
Line 25 an owner-based...
Answer: It was included an as suggested.
Line 26 Manual thigh circumference measurements
Answer: It was included manual as suggested.
Line 27 showed significant increases over the trial period with a greater degree of change observed after week 8.
Answer: It was changed as suggested: “Manual thigh circumference measurements of both hind limbs showed significant increases over the trial period with a greater degree of change observed after week 8.”
Line 28 ... of the left. and the quadricep femoris muscles of both hind limbs showed
significant increases in the cross-sectional thickness post treatment.
Answer: It was changed as suggested: “Ultrasound measurements of the left gluteal muscle and the quadriceps femoris muscles of both hind limbs showed significant increases in the cross-section thickness post WBV.”
Line 31 48 WBV sessions when compared to pre-treatment patterns.
Answer: It was changed: “Surface EMG of the vastus lateralis muscle was closer to normal patterns after 48 WBV sessions when compared to pre-session patterns.
Line 33 Maybe something like this to wrap up. Therefore, further studies will help to
better clarify the role of WBV in canine rehabilitation protocols.
Answer: The phrase was included as suggested.
Line 37 that has an apparently linear progression over time.
Answer: The text was changed as suggested. The word continuous was placed because of the other reviewer. “Canine hip dysplasia is a complex developmental skeletal disease that has an apparently continuous progression over time.”
Line 38 clinical signs of dysplasia
Answer: It was included.
Line 39 tearing
Answer: It was included.
Line 40 of age, associated ....
Answer: It was included.
Liine 41 Even though radiographic signs
Answer: It was included.
Line 42 [2[ the clinical presentation
These are examples of better clarification. Please review the writing to better define your results...
Answer: It was changed.
Line 53 specifically what for as an “alternative method of physiotherapy and rehabilitation”. Make sure you give solid reasons for why the reader should think this is a good therapy and specifically for dysplasia. I don’t think this paragraph does this.
Answer: the text was included: “…, such as exercise modality for improving muscle performance in different types of sports and as an alternative method of physiotherapy and rehabilitation for musculoskeletal, neurological, or metabolic conditions [11,12].
Materiais and Methods
The order could be improved so that one Dogs
Evaluation of body condition Pain and lameness assessments
Blood tests and muscle size evaluations Radiographic findings
Kinetic gait analysis Surface electromyography
Whole-Body Vibration Platform Statistical Analyses (plural not singular)
Answer: We believe that this sequence will be confusing because of the different time points of evaluation for each item.
We reunite Dogs and Radiographic examination
We changed Statistical Analysis for Statistical analyses
I would discuss the differences between the types of WVP to show that you used the best type.
For example:
Linear vibration plates produce vibrations that travel in a straight line, while tri-planar vibration plates produce multidirectional vibrations that travel up and down, side to side, and front to back. Pivotal vibration plates produce an oscillating motion that simulates the motion of walking. Etc
Answer: The text was included: “The vibration characteristics of the machine are one of the factors that may influence the effects of the WBV on the body [13]. For example, linear vibration plates produce vibrations that travel in a straight line, tri-planar vibration plates generate multidirectional vibrations, and pivotal vibration plates produce an oscillating motion [25]. The machine of the present study differs from these because it provides vortex wave circulation with the distribution of pulsatile vibration for all limbs, being well-tolerated by the dogs [17].”
Liine 71 Just say 10 dogs were used in the study and discuss inclusion and exclusion criteria. It is confusing because it appears from this paragraph that 28 dogs were used in the study. Line 192-194 belongs here.
Answer: We really examined 28 dogs. However, only 10 met in inclusion criteria as described in the results.
Line 101 Improve wording, for example. WBV training program consisted of 15-minute sessions 3 times weekly for 16 weeks.
Answer: The text was changed as suggested: WBV training program consisted of 15-minute sessions 3 times weekly (intercalary days) for 16 weeks (total = 48 sessions).
Line 113 Were these lameness scores defined by any veterinary association?
Answer: This lameness scores has been used in several orthopedic papers and it is an excellent book that was cited. The number of reference was changed of position: “Visual lameness scoring at walking and trotting [21] was established….”
Line 120 Blood tests
Answer: The text was changed as suggested: “Blood tests included complete…”
Line 124 were measurements then averaged or what was done with the 3 measurements
Answer: The text was included: “The mean of the measures was used for statistical comparison.”
Line 125 Ultrasound evaluations pre and post must be from the same location and this information is essential to convince the reader of your results.
Answer: The text was included: “All ultrasound measurements in the different evaluation time points were undertaken at the same locations by a single experienced investigator.
Line 170 remove the effect of gravity... Not sure how this calculation is specifically figured out but will assume this is a standardized method
Answer: The text was removed as required.
Line 171 Temporal cut ??
Answer: The text was changed to clarify.
Line 174 do you mean averaging five valid gait cycles
Answer: The text was changed to clarify: At least five valid trials, as determined by gait analysis, were selected.
Line 175 what does determined scores mean? This paragraph needs to be explained better so those who are not EMG specialists can understand how you obtained your data. Much better explanation found in lines 352-363. lncorporate some of that wording.
Answer: All text was changed to clarify.
Line 199-203 This has been discussed already about boarderline and mild if these are considered diagnosed dysplasia. This determination is important to the complete discussion and will influence the entire RESULTS section.
Answer: The text was changed: A total of eight dogs had bilateral hip dysplasia. One dog was borderline in both hip joints (no. 7), and the other was borderline in the right hip joint (no. 6) (Table 2).
Line 275-277 Remove
Answer: We apologize. This was part of the template that was not removed.
Line 278 Be careful of the generalized statements, stick with your specific results.
Answer: The text was changed: “Although these results were preliminary, data suggested that WBV provided a beneficial effect on some muscles and probably pain.”
Line 293 this should be in kinetic evaluation section don’t you think?
Answer: No.
Line 295-301 This paragraph is not tied together well. Hard to follow the reasoning behind the selection of studies to support your observations. Again, no significance was found so that needs to be explained if you can.
Answer: The paragraph was removed due to the criticism.
Line 303-315 Interesting paragraph to rationalize why certain muscles were more effected by vibration. Extensor muscles could be more effected...
Answer: Other studies must be done.
The text was included: “The gluteal muscles are lateral muscles of the pelvis related to flexion and extension of the hip and abduct the limb, while the quadriceps femoris muscle is a cranial muscle of the thigh used to support weight and an extensor of the stifle.”
Line 308-312 doesn’t fit in, it’s too general spend more time formulating this interesting connection between the extensor muscle hypertrophy and vibration.
Answer: The text was included: “Based on these human studies, dogs with hip dysplasia that present lameness, difficulty or reluctance to exercise, low levels of fitness, and muscle wastage will be the most likely to obtain a beneficial effect with WBV.”
Line 319 what type of instrumented analysis??? Kinematic, kinetic??
Answer: The text was changed: “The relationship between lameness score and kinetic methods has shown a good correlation in some studies”
Line 320-323 Not really sure where you are going with this sentence. Too generalized need to use more studies specifically about the parameters you used in your study. There are many studies that show just the opposite. This idea needs to be further developed if you want to convince the reader.
Answer: “The text was changed: “The dogs in the present study showed predominantly hind limb lameness at visual lameness scoring; however, signs of pain were observed in the forelimbs in one dog (no. 3). A study in dogs with hip dysplasia also reported signs of pain in other joints of the hind limbs as well as forelimb joints [7].”
Line 327 decrease in what
Answer: The text was changed to clarify: “On the other hand, the scores obtained by a questionnaire completed by owners about signs of pain, discomfort, and difficulties with locomotion revealed a statistically significant decrease among evaluation time points.”
Line 329 remove “on the other hand”
Answer: It was removed.
Line 339 Really like this paragraph. But issue would be intra subject reliability you can compare pre and post within the same dog to obtain a difference score which could then be evaluated inter subject.
Answer: The last part of paragraph was excluded.
Line 364 Other limitations need to be included here. One major problem is lack of control group.
Answer: It was included “This pilot study has several limitations. First, the absence of a control group; therefore, the results should not be generalized. Unfortunately, owners of healthy dogs did not agree to participate in the study. Second, different dog breeds and small sample size were some inherent limitations associated with a clinical trial. Thus, further studies using a control group, specific breeds with similar radiographic and clinical signs, and a higher number of dogs are mandatory to confirm the role of WBV in hip dysplasia dogs.”
Line 366 suggested is even too strong to be supported by your data Line
Answer: It was changed
369 Your data does not support this statement.
Answer: It was changed
Line 371 Your data supports this statement.
Answer: Thanks.
Conculsions need to be rewritten to support other changes in the paper that are needed.
Answer: We apologize that the same part was inserted two times.
The text was changed as suggested: “The findings of this pilot study suggested that the long-term WBV program could potentially improve hind limb muscle hypotrophy and pain as reported by the owners in lame dogs with hip dysplasia or borderline but with no change in the percentage of limb loadings or visual lameness score. Further studies evaluating the use of WBV for canine hip dysplasia appear to be justified.”
Reviewer 3 Report
Comments and Suggestions for Authors
Comments for author
Thank you for submitting this interesting paper. I have a few questions and comments for the authors.
Comments
1. Line 39-41 and line 110;Regarding the assessment of pain reduction, how did you distinguish between the effect of whole body vibration and pain reduction due to reduced inflammation? Also, since juvenile and mature dogs have different pain mechanisms, shouldn't they be evaluated at different ages?
2. Line127-128; I would like to know why you chose these muscles.
3. Line 130-132; Please explain how the cranial, medium, and caudal of the glutei muscle, and proximal, medium, and distal of other muscle groups were set up.
4. Line 157; I know that the vastus lateralis muscle is easy to measure EMG, but I would like to know why you measured it in the vastus lateralis and not the glutei muscle or other muscles, and the significance of measuring the vastus lateralis muscle in dogs with hip dysplasia.
5. Figure 1 (Line 219); There is a scale up to 60. Did you add up the scores for each of the items and indicate them? If so, please indicate the overall score range and the average score for a normal dog.
6. Figure4 (line 257); Figure not shown correctly. Please indicate which lines are week 0, 8, 16.
7. Table 5; Is there any reason why No 2 dog was not evaluated on the right hind leg?
Introduction and Discussion
8. In the Introduction, the authors list several hypotheses for the mechanism of improvement of whole body vibration that have not been clarified. Was the purpose of this study to evaluate the relevance of these mechanism? I do not understand which mechanism the authors were trying to prove and what method they were trying to use to evaluate it.
Line 353-354; I would like you to explain the normal EMG pattern.
Comments on the Quality of English LanguageI do not think any major problems with the quality of the English in this paper.
Author Response
REVIEWER 3
Comments and Suggestions for Authors
Comments for author
Thank you for submitting this interesting paper. I have a few questions and comments for the authors.
Thank you very much for taking the time and effort to review the manuscript. We sincerely appreciate your positive and constructive feedback.
Comments
- Line 39-41 and line 110;Regarding the assessment of pain reduction, how did you distinguish between the effect of whole body vibration and pain reduction due to reduced inflammation? Also, since juvenile and mature dogs have different pain mechanisms, shouldn't they be evaluated at different ages?
Answer: I think these are good questions. However, other studies using other methodologies should be done to answer these questions.
- Line127-128; I would like to know why you chose these muscles.
Answer: Because ultrasonographic evaluation had already done in most of these muscles (Use of B-mode ultrasonography for measuring femoral muscle thickness in dogs - SAKAEDA and SHIMIZU, 2016). Also, they are important muscles for the cranial thigh (quadriceps femoris), caudal thigh (biceps femoris), and lateral pelvis (gluteal muscles).
The text was included: Significant changes were found in the thigh circumference measurements and ultrasound measurements of the left gluteal muscles and bilateral quadriceps femoris muscle after WBV in the dogs. The gluteal muscles are lateral muscles of the pelvis related to flexion and extension of the hip and abduct the limb, while the quadriceps femoris muscle is a cranial muscle of the thigh used to support weight and an extensor of the stifle.
- Line 130-132; Please explain how the cranial, medium, and caudal of the glutei muscle, and proximal, medium, and distal of other muscle groups were set up.
Answer: Due to the anatomic position of the muscles. The glutei muscle has a different position.
- Line 157; I know that the vastus lateralis muscle is easy to measure EMG, but I would like to know why you measured it in the vastus lateralis and not the glutei muscle or other muscles, and the significance of measuring the vastus lateralis muscle in dogs with hip dysplasia.
Answer: Theoretically, any muscle could be affected by the WBV. As you said the vastus lateralis muscle was used because was easier to evaluate and we also already studied this muscle (Negrão, R.R.; Rahal, S.C.; Kano, W.T.; Mesquita, L.R.; Hormaza, J.M. Analysis of time series of surface electromyography and accelerometry in dogs). This muscle is important to locomotion and is a stifle extensor.
- Figure 1 (Line 219); There is a scale up to 60. Did you add up the scores for each of the items and indicate them? If so, please indicate the overall score range and the average score for a normal dog
Answer: The figure was changed as required. Normal dog the score is 0.
- Figure4 (line 257); Figure not shown correctly. Please indicate which lines are week 0, 8, 16.
Answer: It was corrected: “Figure 4. Surface EMG signs of muscle activity of the vastus lateralis muscle (a – left hind limb; b – right hind limb) during stance phase and swing phase in a dog with hip dysplasia, before (t1), and at weeks 8 (t2)and 16 (t3) of Whole-Body Vibration.”
- Table 5; Is there any reason why No 2 dog was not evaluated on the right hind leg?
Answer: The information was included in the table: insufficient quality for use of the surface EMG signals)
Introduction and Discussion
- In the Introduction, the authors list several hypotheses for the mechanism of improvement of whole body vibration that have not been clarified. Was the purpose of this study to evaluate the relevance of these mechanism? I do not understand which mechanism the authors were trying to prove and what method they were trying to use to evaluate it.
Answer: The information was to show to the reader that the action of the WBV is too complex.
Line 353-354; I would like you to explain the normal EMG pattern.
Answer: All methodology used for sEMG was included in the text to clarify
Comments on the Quality of English Language
I do not think any major problems with the quality of the English in this paper
Answer: Thanks.
Round 2
Reviewer 1 Report
Comments and Suggestions for Authors
Dear authors,
Thank for the work done addressing the comments made. Still, there are points I feel need to be corrected.
Please find specific comments below.
You can use book chapthers, but you should perfer papers, as they are peer-reviewed, regardless of the quality of the bokd. You should be able to find papers covering the topics in which you cited books.
Regarding the BMI, the final value is given in kg/m2. From my point of view, when presented as “body weight (kg)/height (m2 )”, it seems that body weight was measured in Kg and height in m2. It should read “body weight/height (kg/m2)”.
You wrote “Owner-based questionnaires were used to evaluate pain intensity, as previously described for dogs with hip dysplasia [20].” This suggests that several questionnaires were used. You only presente information for one. Which one was used? Is this questionnaire validated? Based on the reference provided, the questionaire was in Finnish. How was it translated for Portuguese? This is very importante, as there is a need for using validated tools, otherwise the results are not reliable.
The same can be said of the methodology for thigh circumference evaluation.
“The hips were not evaluated post-treatment. The results were separated because the data could influence the kinematic data.” I need more clarification on this information, as I feel the original comment “OFA scores are usually given for the combined hips” was not well addressed.
Comments on the Quality of English LanguageNo comments.
Author Response
REVIEWER 1
Thank you again for taking the time and effort to review the manuscript. We sincerely appreciate your positive and constructive feedback.
Thank for the work done addressing the comments made. Still, there are points I feel need to be corrected.
Please find specific comments below.
You can use book chapthers, but you should perfer papers, as they are peer-reviewed, regardless of the quality of the bokd. You should be able to find papers covering the topics in which you cited books.
Answer: I changed two, but other I could not find.
Regarding the BMI, the final value is given in kg/m2. From my point of view, when presented as “body weight (kg)/height (m2 )”, it seems that body weight was measured in Kg and height in m2. It should read “body weight/height (kg/m2)”.
Answer: I apologize for the mistake. The formula was corrected: body weight (kg)/height (m)2.
You wrote “Owner-based questionnaires were used to evaluate pain intensity, as previously described for dogs with hip dysplasia [20].” This suggests that several questionnaires were used. You only presente information for one. Which one was used? Is this questionnaire validated? Based on the reference provided, the questionaire was in Finnish. How was it translated for Portuguese? This is very importante, as there is a need for using validated tools, otherwise the results are not reliable.
Answer: An owner-based questionnaire was used to evaluate pain intensity based on one previously described for dogs with hip dysplasia and translated into Portuguese.
We did not have when you made this project a questionnaire validated to Portuguese.
The text was included: An owner-based questionnaire was used to evaluate pain intensity based on one previously described for dogs with hip dysplasia and translated into Portuguese [20]. The questions answered by the owner’s dog had a descriptive scale of 0, 1, 2, 3, or 4. Each item was scored from 0 (the best) to 4 (the worst).
The same can be said of the methodology for thigh circumference evaluation.
Answer: We use the same methodology in all evaluation time points. There other published studies that used this same methodology.
“The hips were not evaluated post-treatment. The results were separated because the data could influence the kinematic data.” I need more clarification on this information, as I feel the original comment “OFA scores are usually given for the combined hips” was not well addressed.
Answer: In our study, each hip joint must assessed independently in the same ventrodorsal hip-extended radiographic projections of the pelvis and scored since this data influences the clinical evaluation (lameness in each limb), and kinetic gait analysis in each limb.
Reviewer 2 Report
Comments and Suggestions for Authors
Author Response
REVIEWER 2
Thank you again for taking the time and effort to review the manuscript. We sincerely appreciate your positive and constructive feedback.
There are many grammar errors which I would assume will be caught by the final editor. Only the errors that effect the understanding of the material have been included in this review.
I have spent many hours reviewing this paper because there are many interesting points. I understand that the English can be difficult and have tried to be of assistance but did not correct every issue. There may be a few places where the same change must be made in several parts of the paper to maintain consistency. I am just running out of time to offer further detail so please review the information below and apply to all parts of the paper when applicable even if the specific sentences are not noted.
Answer: We appreciate very much your help to improve the manuscript.
Simple Summary: Whole-body Vibration (WBV) on vibrating platforms has been used as an alternative method of physiotherapy and rehabilitation for musculoskeletal, neurological or metabolic conditions in humans. However, in dogs, the use….
Answer: The text was changed as required.
Line 21. This statement is correct however later in the paper the statement is made “dogs with hip dysplasia or boarderline” which is confusing. All places in the paper where this statement is made must be replaced with “dogs with boarderline to severe hip dysplasia”
Answer: The text was changed as required.
Line 23 the left gluteal
Answer: The text was changed as required.
Line 31 cross-sectional
Answer: The text was changed as required.
Line 53…observed in dogs with boarderline to severe hip dysplasia.
Answer: The text was changed as required.
Line 62 Sentence not clear. Change as in Simple Summary
Answer: The text was changed ´Whole-body Vibration (WBV) on vibrating platforms has been used for improving muscle performance in different types of sports and as an alternative method of physiotherapy and rehabilitation for musculoskeletal, neurological or metabolic conditions in humans [11,12].
Line 77 change boarderline
Answer: The text was changed as required.
Line 103 how many dogs had forelimb radiographs?
Answer: This was included in the results: “Radiographs of the forelimbs were done in one dog (no. 3) with pain on elbow palpation. Mild signs of osteoarthritis were verified.”
Line 118 leave out intercalary days not the definition you want. If you want to put the actual days or just leave it at 3 times weekly
Answer: The text was changed as required: “….15-minute sessions 3 times weekly for 16 weeks (total = 48 sessions).”
Line 131 A well-established lameness evaluation scoring system [21] was utilized at the walk and trot by an experienced….
Answer: The text was changed as required.
Line 140 Blood tests and muscle size evaluations
Answer: The text was changed as required.
Line 145 The mean was calculated from the 3 measurements taken and this mean value was used for the statistical calculations.
Answer: The text was changed as required.
Line 154 anatomical landmarks
Answer: The text was changed as required.
Line 159 All pre and post treatment ultrasound measurements were taken utilizing boney landmarks to ensure that the images were taken at the same location.
Answer: The text was changed to attend to the other reviewer as well: All pre-and post-treatment ultrasound measurements were taken by a single experienced investigator utilizing bony landmarks to ensure that the images were taken at the same location.
Line 162 before WBV and at weeks 4, 8, 12 and 16 during the WBV protocol.
Answer: The text was changed as required.
Line 190 A Figure of the placement of the sensors would be helpful
Answer: We used exactly as described by Bockstahler et al. (2012), which showed a figure about this.
Line 196 and a mean value was obtained?? What happened to these 5 valid trials?
Answer: The text was changed to clarify: The mean of five valid trials, as determined by gait analysis, was selected.
I am not a EMG expert so I am assuming all this information is correct and accurate. Please carefully read to make sure this section has no mistakes.
Answer: Two physicists made this part.
Line 252 Was statistical analyses performed on EMG signal changes to find significant differences. This needs to be clarified. If no statistical tests were performed then you need to justify why you determined there were differences in the signals over time. There are several areas of the paper that this needs to be clarified.
Answer: The statistical analyses was not done. The results were included to explain
Line 253 blood tests not laboratory tests, need to be specific. Also manual thigh circumference measurements
Answer: The text was changed as required.
Line 257 R Core Team, R: A Language and Environment for Statistical Computing
Answer: The text was changed as required. Some corrections were done in the statistical analysis.
Line 266 had mild to severe bilateral hip dysplasia.
Answer: The text was changed as required.
Line 278 …signs of osteoarthritis which was more accentuated in the lefi
Answer: The text was changed as required.
Line 281 New paragraph The subjective scores….
Answer: The paragraph was done.
Line 288 Blood tests
Answer: The text was changed as required.
Line 306 not clear … were observed between dogs for the percentage of body weight distribution and the metacarpal/metatarsal pad, toes and paw contact areas of each limb
Answer: The text was changed as required.
Line 311…of muscle activity, as determined by surface EMG, in the….
Answer: The text was changed as required.
Line 314 increase in the percentage of muscle activity…when?
Answer: The beginning of the text showed when: By week 16 (after 48 sessions of WBV), there was an increase in the percentage of muscle activity, and the peak shifted to the middle third of the stance phase.
The text was changed: At week 16….
Line 315 This shift towards a more normalized pattern…to what specifically? To the middle third of the towards a stance phase?
Answer: The text was changed to clarify: At week 16 (after 48 sessions of WBV), there was an increase in the percentage of muscle activity, and the peak shifted to the middle third of the stance phase. Therefore, based on graphs of the EMG activity patterns of the vastus lateralis muscle, an improvement was verified after 48 WBV sessions when compared to pre-session patterns in both hind limbs for five dogs (nos. 1, 2, 4, 6, 7) and in the left hind limb for three dogs (nos. 3, 5, 9) (Table 6).
Line 317 improved how? You need to be very specific here so the reader can understand the changes in the EMG pattern you observed. Please look over this entire paragraph and make it as clear as possible what you were observing and what specifically changed during the treatment protocol.
Answer: The text was changed to clarify: “Therefore, based on graphs of the EMG activity patterns of the vastus lateralis muscle, an improvement was verified after 48 WBV sessions when compared to pre-session patterns in both hind limbs for five dogs (nos. 1, 2, 4, 6, 7) and in the left hind limb for three dogs (nos. 3, 5, 9) (Table 5).”
Line 300 p=0.00 ??
Answer: 0.001. We sent the data to another statistical professor, who reevaluated the statistical data.
Discussion section overall must have a summarizing sentence at the end of each paragraph to explain how your results are important to the advancement of the scientific knowledge you are presenting.
Line 468- 472 This information needs to be in the Materials/Methods section. It can also be referred to here as part of the Discussion but must be in the section that describes the specifics of the tools used in Materials and Methods.
Answer: The focus of the study is not compare the TheraPlate with other machines.
For example, linear vibration plates produce vibrations that travel in a straight line, tri-planar vibration plates generate multidirectional vibrations, and pivotal vibration plates produce an oscillating motion [25]. The machine used in the present study differs from these because it provides vortex wave circulation which allows the pulsatile vibration to be equally distributed to all 4 limbs, which is well-tolerated by the dogs [17].
Answer: The text was changed: The machine used in the present study (TheraPlate) differs from these because it provides vortex wave circulation that allows the pulsatile vibration to be equally distributed to all four limbs, which is well-tolerated by the dogs [17].
I would think this difference in vibration plate is a large issue with the outcome, but you do not state this. Have there been studies showing differences in outcome due to the vibration plate utilized? If so, this could be used as a reason to negate your comparisons to other studies, due to the fact that your outcome is only due to the difference in the vibration plate you used. You need to address this.
Answer: Again, the focus of the present study was not compare the data with other machines. The difference of our study with others using the Theraplate is only the number of sessions. For this, the title of our study is long-term effects.
The text was changed to clarify: “The total number of sessions used in the present study was higher than that reported by other studies in dogs using the TheraPlate [16,17]. However, the vibration platform used in those studies as well as the frequency and duration of each WBV, was the same [16,17].”
Line 474 Doesn’t make sense “despite the frequency and duration of each WBV session as well as the device had been the same [16,17]. “Reword to: …other studies in dogs. However, the vibration platform used in those studies as well as the frequency and duration of each WBV session, was the same.
Answer: The text was changed as required.
But, again your vibration plate was different than the plate used in other studies (at least that is what it appears from your paper) You need to address if this difference effects your results.
Answer: No. The difference was only the total number of sessions as cited: “The total number of sessions used in the present study was higher than that reported by other studies in dogs.”
Line 479 Make a conclusion for this paragraph. This is what discussion is for…. Such as…Therefore, the increased number of sessions which was utilized in the protocol for this study may not be necessary and the number of sessions may be able to be reduced in future studies if only CBC and creatine kinase levels are evaluated.
Answer: No, this is not the idea of the study. It was included: Therefore, the protocol used was considered safe and feasible for the objective of this study.
Line 478 ‘ïn” replaced with ”for” as well as in all other spots where this statement was used for all evaluation time points.
Answer: It was changed as required.
Line 481 of the dogs in this study.
Answer: It was changed as required.
Line 482 In addition, some dogs
Answer: It was changed as required.
Line 488 no significant change occurred ? Or just based on descriptive statistics? change occurred in what?
Answer: It text changed as required: “…influence the results since no significant change occurred for all evaluation time points.”
But you state that one of the limitations of the study was that different breeds were used, but here you say there was no indication that body mass and body weight showed different results…. clarify these discrepancies
Answer: I think you did not understand. If the body weight and the body mass index changed during the study the data could be altered. There are studies showing that body weight reduction may cause a decrease in lameness in dogs with osteoarthritis.
Line 488 If you leave this sentence here then you must tie this to the specific discussion… body mass
index did not influence the radiographic findings, lameness scores or orthopedic evaluation since no change occurred in either of these parameters during the study. This statement can be added to any of the findings where it is applicable so you must decide when it is appropriate to discuss and when it is not.
Answer: The text was changed: “….did not influence the lameness score since no significant change occurred for all evaluation time points.”
Line 489. What is the conclusion of this paragraph? Must have a summary sentence.
Answer: The text was changed to clarify: In addition, it is worth mentioning that the body weight and the body mass index did not influence the lameness score since no significant change occurred for all evaluation time points.
Line 494 why is your device considered not as accurate? Is there a citation? Or is it just a clinical generalization? State the reason there may be a disadvantage to your technique.
Answer: The text was remove due to the criticism.
Line 495 and abduction of the hind limb.
Answer: It was changed as required
Liine 496 of the stifle while also flexing the hip.
Answer: The text was changed as required
Line 497 In addition, human studies have shown that WBV reduced
Answer: The text was changed as required: “In addition, human studies have shown that WBV reduced atrophy in the knee and ankle extensors muscles in patients…..
Summarize this information in a sentence at the end of this paragraph about how interesting it is that the extensor muscles may be more effected than other muscles by the WBV protocol and why you think this might be true. Also discuss why the abductor muscles may be more influenced. These are you statistically significant findings so focus on this info. Super interesting.
Answer: The text was included: “More studies are needed to ascertain the response in each thigh muscle since whole-body vibration revealed statistically significant positive effects in two muscles but no difference in sartorius, vastus, and biceps femoris muscles on ultrasound estimation.”
MUST ADD A TABLE SHOWING THE MEASUREMENTS OF THE MUSCLES EVALUATED BY ULTRASOUND even though you have the measurements in the text
Answer: The table was included and the text was changed.
Line 498 New paragraph
Answer: It was done.
Line 500… a review of literature of 27 human studies of the use of WVB in the aging in the aging population, 9 of the studies evaluated muscular strength and all studies found significant? improvement in (specifically what?)
Line 501…. In 9 of the studies muscular strength of … (which muscles?)
Answer: The text was changed to clarify: On the other hand, a systematic review of the use of WBV in the aging population identified nine studies that observed statistically significant improvements in lower-body muscular strength [31].
Line 503 You did not specifically discuss “lameness”or “difficulty or reluctance to exercise” as being evaluated in your cited literature. Therefor, these specific improvements cannot be stated as classifications that will most likely benefit from your protocol. You cannot make assumptions from the cited literature that are not stated in the cited literature.
Answer: The text was changed: “…..dogs with hip dysplasia that present low levels of fitness, and muscle wastage will be the most likely to obtain a beneficial effect with WBV.”
Line 505 statistically significant positive results. Rewite this last sentence because some of this information will now be in the previous paragraph summary sentence.
Answer: The text was changed: “More studies are needed to ascertain the response in each thigh muscle since whole-body vibration revealed statistically significant positive effects in two muscles but no difference in sartorius, vastus, and biceps femoris muscles on ultrasound estimation.”
Line 512 kinetic evaluation
Answer: It was changed as required
Line 520 might be influenced by the placebo effect
Answer: It was changed as required
Line 521 For example, owners may perceive improvement in the dogs due to other issues than just an improvement in lameness, such as the dogs feel happier and more comfortable. While these characteristics are subjective, they can reflect an improvement in the quality of life of these dogs due to the treatment.
Answer: It was changed as required
533 it was observed
Answer: It was changed as required\
- for the lame limb….than for the…
Answer: It was changed as required: ”It was observed that the paw contact area was smaller for the lame limb than for the contralateral…”
Similarly, one study found that alterations…changes in paw contact area [34]. Therefore, a…contact area values which was also found in this study.
Answer: It was changed as required.
537 lame limb
Answer: It was changed as required.
- closer to normal patterns When? at week 16?
Answer: The text was changed to clarify: However, based on graphs of the EMG activity patterns of the vastus lateralis muscle, 65% of the hind limbs had an improvement after 48 WBV sessions when compared to pre-session patterns.
548 In addition, 2 studies
Answer: It was changed as required.
551 Summarize what this means….Therefore, it appears that the EMG signal showed a change in pattern from those observed in lame dogs to patterns observed in sound dogs in 50% of the dogs after the use of WBV for 16 weeks.
Answer: It was included as suggested.
Conclusions
The findings of this pilot study in dogs with boarderline to severe hip dysplasia suggested that a long- term WBV program could potentially improve hind limb muscle hypotrophy in some muscles and owner-reported pain in lame dogs. However, no significant changes were found in the percentage of limb loadings or for the visual lameness scores when pre-treatment evaluations were compared to post- treatment evaluations. Further studies evaluating the use of WBV for canine hip dysplasia appears to be justified.
Answer: It was changed as required.
Figure 2 Must have measurement units. Must state these are the manual thigh circumference measurements
Answer: It was corrected as required: Box plot of the manual thigh circumference measurements (cm) in left”..”
Figure 4 Include which dog this is based on your dog numbers
Answer: number 4
Table 1 must indicate what numbers mean for BCS in Table info
Answer: It was corrected: “….body weight, 9- body condition score (BCS), and..”
Table 2 must have clarification of what the numbers mean for each category in the Table info. For example, Radiographic hip score: 0=no dysplatic categories; 1=borderline; 2=mild; 3=moderate; 4=severe.
Lameness score…etc…
Answer: It was included:
Radiographic hip score: 0=no; dysplatic categories, 1=borderline, 2=mild, 3=moderate, 4=severe.
Lameness score: 0=no; lameness with weight-bearing, 1=slight; 2=obvious; 3=severe; lameness non-weight-bearing, 4=intermittent; 5 – continuous.
Othopedic examination (pain, crepitus, decreased range of motion in the hip joint): 1=absent, 2=mild, 3=moderate, 4=severe.
Table 4 List measurement units…. For example mm or cm or m
Answer: This information already is the table: Paws (cm²), Pads (metacarpus/metatarsus) (cm²), Toes (cm²)
Table 5 Were these changes evaluated statistically to show significant differences? If not then you must state that the evaluations were subjective based on clinical observation.
Answer: The text was changed as required: “Table 5. Subjective evaluation of the surface EMG activity patterns before…”
Reviewer 3 Report
Comments and Suggestions for Authors
Thank you for submitting your revised manuscript. However, there are still a few questions I would like you to answer.
- Line 39-41 and line 110;Regarding the assessment of pain reduction, how did you distinguish between the effect of whole body vibration and pain reduction due to reduced inflammation? Also, since juvenile and mature dogs have different pain mechanisms, shouldn't they be evaluated at different ages?
Answer: I think these are good questions. However, other studies using other methodologies should be done to answer these questions.
Comment; Please state this issue as a limitation and explain what study needs to be done on these questions using other methodologies in the future.
- Figure 1 (Line 219); There is a scale up to 60. Did you add up the scores for each of the items and indicate them? If so, please indicate the overall score range and the average score for a normal dog
Answer: The figure was changed as required. Normal dog the score is 0.
Comment: Please describe how the pain score values are caluculated. Because the pain score for reference 20 is from 0 to 4.
- In the Introduction, the authors list several hypotheses for the mechanism of improvement of whole body vibration that have not been clarified. Was the purpose of this study to evaluate the relevance of these mechanism? I do not understand which mechanism the authors were trying to prove and what method they were trying to use to evaluate it.
Answer: The information was to show to the reader that the action of the WBV is too complex.
Comment: Is this study simply a report of results using WBV? Did the authors not conduct this study to prove some effect of action? The mechanism of action is complex, but explain what the authors expected WBV to do in that context. If you did not focus on the mechanism of action, discuss how you can prove the mechanism that led to this result.
9. Line 273; It is better to match the title with the title of line 129 (materials and methods) and to list the pain results first.
Author Response
REVIEWER 3
Thank you again for taking the time and effort to review the manuscript. We sincerely appreciate your positive and constructive feedback.
Thank you for submitting your revised manuscript. However, there are still a few questions I would like you to answer.
Line 39-41 and line 110;Regarding the assessment of pain reduction, how did you distinguish between the effect of whole body vibration and pain reduction due to reduced inflammation? Also, since juvenile and mature dogs have different pain mechanisms, shouldn't they be evaluated at different ages?
Answer: I think these are good questions. However, other studies using other methodologies should be done to answer these questions.
Comment; Please state this issue as a limitation and explain what study needs to be done on these questions using other methodologies in the future.
Answer: The limitations were included as required: “In addition, studies using other methodologies will be necessary to understand the role of WBV in pain reduction and/or inflammation and the different pain mechanisms involved in juvenile and mature dogs.”
- Figure 1 (Line 219); There is a scale up to 60. Did you add up the scores for each of the items and indicate them? If so, please indicate the overall score range and the average score for a normal dog
Answer: The figure was changed as required. Normal dog the score is 0.
Comment: Please describe how the pain score values are caluculated. Because the pain score for reference 20 is from 0 to 4.
Answer: The text was included to clarify: “An owner-based questionnaire was used to evaluate pain intensity based on one previously described for dogs with hip dysplasia and translated into Portuguese [20]. The questions answered by the owner’s dog had a descriptive scale of 0, 1, 2, 3, or 4. Each item was scored from 0 (the best) to 4 (the worst).”
- In the Introduction, the authors list several hypotheses for the mechanism of improvement of whole body vibration that have not been clarified. Was the purpose of this study to evaluate the relevance of these mechanism? I do not understand which mechanism the authors were trying to prove and what method they were trying to use to evaluate it.
Answer: The information was to show to the reader that the action of the WBV is too complex.
Comment: Is this study simply a report of results using WBV? Did the authors not conduct this study to prove some effect of action? The mechanism of action is complex, but explain what the authors expected WBV to do in that context. If you did not focus on the mechanism of action, discuss how you can prove the mechanism that led to this result.
Answer: The text was changed as required: The mechanism by which WBV may enhance performance and muscular strength has not been well understood [12,13].
- Line 273; It is better to match the title with the title of line 129 (materials and methods) and to list the pain results first.
Answer: The text was changed as required.